# Compact Physics Hot-Carrier Degradation Model Valid over a Wide Bias Range

**DOI:** 10.3390/mi14112018

**Published:** 2023-10-30

**Authors:** Stanislav Tyaginov, Erik Bury, Alexander Grill, Zhuoqing Yu, Alexander Makarov, An De Keersgieter, Mikhail Vexler, Michiel Vandemaele, Runsheng Wang, Alessio Spessot, Adrian Chasin, Ben Kaczer

**Affiliations:** 1Imec, Kapeldreef 75, 3001 Leuven, Belgium; erik.bury@imec.be (E.B.); alexander.grill@imec.be (A.G.); alexander.makarov@imec.be (A.M.); an.dekeersgieter@imec.be (A.D.K.); alessio.spessot@imec.be (A.S.); adrian.chasin@imec.be (A.C.); ben.kaczer@imec.be (B.K.); 2Institute of Microelectronics, Peking University, Beijing 100871, China; zhuoqing_yu@163.com (Z.Y.); r.wang@pku.edu.cn (R.W.); 3A.F. Ioffe Institute, Polytechnicheskaya 26, 194021 St.-Petersburg, Russia; vexler@mail.ioffe.ru

**Keywords:** hot-carrier degradation, compact physics model, secondary carriers, impact ionization, interface traps, carrier transport

## Abstract

We develop a compact physics model for hot-carrier degradation (HCD) that is valid over a wide range of gate and drain voltages ($V_{\mathrm{gs}}$ and $V_{\mathrm{ds}}$, respectively). Special attention is paid to the contribution of secondary carriers (generated by impact ionization) to HCD, which was shown to be significant under stress conditions with low $V_{\mathrm{gs}}$ and relatively high $V_{\mathrm{ds}}$. Implementation of this contribution is based on refined modeling of carrier transport for both primary and secondary carriers. To validate the model, we employ foundry-quality n-channel transistors and a broad range of stress voltages $\left\{V_{\mathrm{gs}},V_{\mathrm{ds}}\right\}$.

## 1. Introduction

Hot-carrier degradation (HCD) has been recognized as the most harmful degradation issue limiting the lifetime of modern metal-oxide semiconductor field-effect transistors (MOSFETs) [1,2,3]. As such, comprehensive and predictive modeling of HCD is crucial for enabling further development of micro/nanoelectronics. Due to the complexity of the physical mechanisms underlying HCD [4,5], available physics-based models for HCD are computationally expensive [6,7,8,9,10,11]. On the other hand, empirical and phenomenological models [12,13,14,15,16,17,18,19,20,21] lack predictive capabilities because they do not capture the entire physical picture behind HCD. Hence, they cannot ensure that device lifetime under the operating regime is adequately predicted based on available experimental data acquired under more aggressive stress conditions, with most probably another dominant mechanism driving HCD.

In order to reach a compromise between model accuracy and optimized computational resources, we recently developed a compact physics model (CPM) for HCD [22], which was implemented on top of the reliability simulator Comphy [23]. This CPM relied on a simplified description of carrier transport for primary carriers and was shown to capture HCD in short-channel FETs stressed under the worst-case conditions (WCC) of HCD, i.e., under $V_{\mathrm{gs}}$ = $V_{\mathrm{ds}}$ ($V_{\mathrm{gs}}$ and $V_{\mathrm{ds}}$ are gate and drain voltages, respectively) [24,25,26,27]. However, the model considered only the contribution of primary carriers to HCD, and the contribution of secondary carriers generated by impact ionization (II) was not implemented. On the other hand, in our recent publications, we demonstrated both experimentally [28] and theoretically [29,30] that under HC conditions with $V_{\mathrm{gs}}$ substantially lower than $V_{\mathrm{ds}}$, secondary carriers provide a strong contribution to HCD. Moreover, secondary carriers generated by impact ionization were shown to give rise to the so-called turn-around effect when contributions to total damage related to primary and secondary carriers partially compensate each other. For example, screening of damage produced by primary electrons by the degradation component driven by secondary holes in lightly-doped drain nMOSFETs was reported by Vuillaume et al. [31]. Next, Chen et al. [32] demonstrated the same phenomenon in high-voltage drain-extended metal-oxide-semiconductor transistors subjected to hot-carrier degradation. Furthermore, Starkov et al. [33] performed an analysis of the turn-around effect in planar 5 V nMOSFETs based on results obtained with the charge-pumping technique; in a more recent paper [34], they carried out modeling of this phenomenon. Such an intricate behavior (when primary and secondary carriers generate/populate traps located in different device sections) was shown by various groups to be typical also for OFF-state stress [35,36,37]. Finally, in our recent work, we demonstrated that II can be the reason for the stimulated recovery of bias temperature instability (BTI) induced by HCD [38]. Therefore, the main goal of this work is to extend our CPM for HCD by incorporating the impact of secondary carriers on HCD.

Another important improvement of the CPM presented in this paper is the refinement of carrier transport treatment. Indeed, in the previous version of our CPM, average carrier energy was evaluated via the homogeneous energy balance equation used in drift-diffusion (DD) models [39], i.e., this energy was determined by the square of the electric field and the carrier mobility. However, the DD approach to the Boltzmann transport equation (BTE) solution is known to fail to model carrier transport in ultra-scaled FETs [40,41]. Therefore, implementation of the contribution of secondary carriers should rely on refined carrier transport treatment for both primary and secondary carriers [42]. The extended CPM is validated here against HCD data over a broad $\{V_{\mathrm{gs}}$,$V_{\mathrm{ds}}\}$ range.

## 2. Experimental

To validate the model, we used planar n-channel MOSFETs (with primary and secondary carriers being electrons and holes, respectively). Note that we intentionally used planar MOSFETs with simplified geometry. The reason behind this is that transistors of novel architectures—such as fin [43,44], nanowire [45,46], nanosheet [47,48], forksheet [49,50,51], and complementary FETs [52,53]—have confined channels. As a consequence, modeling HCD in such devices would result in additional challenges due to quantum confinement effects and the 3D nature of the FET structure. In this study, however, we focus on the CPM for HCD and strive to minimize the complexity originating from “side effects”. Employed transistors are foundry-quality devices with a channel length of $L_{g}$ = 28 nm and an operating voltage of $V_{\mathrm{dd}}$ = 1.2 V. Their high-*k* gate stack is made of silica and hafnia layers with an equivalent oxide thickness of 1.3 nm. The devices were stressed under the worst-case conditions (WCC) for HCD in short-channel MOSFETs, i.e., at $V_{\mathrm{gs}}$ equal to $V_{\mathrm{ds}}$; for both voltages, we used values of 1.8, 1.9, and 2.0 V. We also subjected these MOSFETs to HC stress at much lower $V_{\mathrm{gs}}$, namely at $V_{\mathrm{gs}}$ of 1.0 V ($V_{\mathrm{ds}}$ was chosen to be equal to 1.9, 2.0, and 2.1 V) and $V_{\mathrm{gs}}$ = 0.69 V ($V_{\mathrm{ds}}$ values were equal to 1.8, 1.9, and 2.0 V). All experiments were conducted at room temperature with stress times of up to 144 s.

To assess HCD, we monitored relative changes ($\Delta I_{d,\mathrm{lin}}$) of the drain current in the linear regime (with $V_{\mathrm{ds}}$ = 50 mV and $V_{\mathrm{gs}}$ = 1.2 V) as a function of stress time (*t*). Recorded $\Delta I_{d,\mathrm{lin}}$ values were relative, i.e., normalized to the drain current in the pristine MOSFET. To enable fast measurements of many samples in parallel we used on-chip smart arrays [54,55]. Consequently, for each combination of $V_{\mathrm{gs}}$ and $V_{\mathrm{ds}}$ we employed ∼3800 samples, obtained $\Delta I_{d,\mathrm{lin}}$ changes, and then for each stress time step, we extracted their mean values. Further in the paper under $\Delta I_{d,\mathrm{lin}}$, we understand these mean values, which are summarized in Figure 1, Figure 2 and Figure 3. Our compact physics model was verified in order to reproduce these mean $\Delta I_{d,\mathrm{lin}}\left(t\right)$ traces.

Note that the extraction of $\Delta I_{d,\mathrm{lin}}$ drifts was based on measurements of entire $I_{d}-V_{\mathrm{gs}}$ curves with the gate voltage sweeping from 0 to 1.2 V (meanwhile, the stress phase was interrupted). Such a procedure required a measurement time of ∼0.75 s, and therefore HCD was assessed with the corresponding delay. Although recovery of HCD (or more precisely, passivation of P${}_{b}$ centers that were created by the rupture of Si-H bonds [56,57,58,59]) was reported by several groups [60,61,62,63,64], this process was shown to have a significant rate only at temperatures of 150 °C or higher. As our experiments were conducted at room temperature, we can conclude that $\Delta I_{d,\mathrm{lin}}$ values did not recover during the aforementioned measurement delay. Another recoverable contribution to the entire damage can originate from the trapping of carriers by defects in the dielectric layer (bias temperature instability) [65,66]. However, this type of degradation is known to be homogeneously distributed over the coordinates along the $\mathrm{Si}/{\mathrm{SiO}}_{2}$ interface. Our recent experimental studies have shown that under the same stress conditions, as in this paper applied to the same devices, the factor of degradation localization is within the range of 0.6–0.8 (a value of this factor equal to 1 corresponds to strong damage localization near the drain). In other words, this type of damage is relatively strongly localized, and therefore the contribution of bias temperature instability can be neglected [55]. The fact that this localization factor is less than 1 stems from the contribution to HCD provided by secondary holes with the position of the corresponding interface trap density maximum shifted towards the source, as compared with the near-drain $N_{\mathrm{it}}$ maximum related to primary electrons (see Section 4). Moreover, our experience in the field of BTI suggests that significant BTI recovery occurs after relaxation time, which is an order of magnitude longer than stress time [67]. This is not the case for our measurements because relaxation time was ∼0.75 s, but the shortest stress duration was 1 s. Hence, even though BTI provides a non-negligible contribution to the total damage, one can neglect the recovery of $\Delta I_{d,\mathrm{lin}}$ values.

In our study, we used very high stress voltages, $V_{\mathrm{gs}}$ and $V_{\mathrm{ds}}$. There are several reasons for this. According to our understanding, the physical mechanism behind HCD is the dissociation of Si-H bonds at the $\mathrm{Si}/{\mathrm{SiO}}_{2}$ interface induced by channel carriers [6,27,68]. The bond dissociation reaction has two pathways, i.e., the single- and multiple-carrier (SC and MC, respectively) mechanisms of bond dissociation [4,69]. Although the MC-process is considered dominant for HCD under low-voltage stress conditions, it was shown that this mechanism can lead to a significant contribution to HCD even in high-voltage transistors [7,70,71]. The MC-process is driven by colder carriers, and a high carrier concentration typically results in a high rate of this mechanism. Therefore, we intentionally used high $V_{\mathrm{gs}}$ values to ensure that this process has a significant rate. As for the SC-process, it is driven by hot carriers whose energies are determined by the applied $V_{\mathrm{ds}}$. To ensure that the SC-mechanism has a high rate as well, we applied large source-drain voltages. One of the goals of this study was to analyze the contribution of secondary carriers generated by II to HCD, and therefore high $V_{\mathrm{ds}}$ values were chosen to enhance this contribution. Finally, in our experiments, stress times were limited by 144 s, and therefore these $\{V_{\mathrm{ds}}$ and $V_{\mathrm{gs}}\}$ values were supposed to result in significant $\Delta I_{d,\mathrm{lin}}$ changes within the aforementioned stress time window.

Let us mention that the time exponents featured by the measured $\Delta I_{d,\mathrm{lin}}\left(t\right)$ curves (Figure 1, Figure 2 and Figure 3) are within the range of 0.2–0.35 and therefore smaller than those reported for HCD by several other groups [72,73]. This is because in our study we used quite aggressive HC stress: one can see that under the lowest stress voltages of $V_{\mathrm{gs}}$ = 0.69 V and $V_{\mathrm{ds}}$ = 1.8 V (see Figure 2), already at a stress time of ∼1 s the $\Delta I_{d,\mathrm{lin}}$ value is ∼2%, while under $V_{\mathrm{gs}}$ = $V_{\mathrm{ds}}$ = 2.0 V (Figure 1), the drain current change $\Delta I_{d,\mathrm{lin}}$ for t∼1s substantially exceeds 10%. We analyzed the behavior of HCD under high stress voltages in one of our previous papers [74]. It was shown that even at short stress times, the drain area of the transistor is already heavily degraded and the concentration of $N_{\mathrm{it}}$ is saturated, i.e., the available Si-H bonds are predominantly broken, and the near-drain $N_{\mathrm{it}}$ value does not vary with the coordinate along the interface. In this scenario, the further increase in $\Delta I_{d,\mathrm{lin}}$ with *t* is due to propagation of the $N_{\mathrm{it}}$ front inside the device channel. As a result, the time exponent of the $\Delta I_{d,\mathrm{lin}}\left(t\right)$ curves is somewhat lower than that typical for milder HCD. More severe stress conditions with a higher $V_{\mathrm{gs}}$ value result in a broader degraded region with almost constant $N_{\mathrm{it}}$ near the device drain, a more saturated HCD, and consequently a smaller time exponent. Such a behavior is consistent with HCD data published by Varghese et al. [75] and Yamagata et al. [76], where the authors have shown that the time slope of degradation traces reduces at higher stress voltages; the same trend was obtained within our TCAD model applied to HCD modeling in finFETs [77]. To conclude, we intentionally used very aggressive stress conditions and therefore our obtained $\Delta I_{d,\mathrm{lin}}\left(t\right)$ dependencies were more gradual compared with those typically monitored during HCD.

## 3. The Model

Our CPM is based on the detailed physical picture underlying HCD, which was captured in the TCAD version of our HCD model [9,78]. Both versions of the model consider the dissociation of Si-H bonds at the $\mathrm{Si}/{\mathrm{SiO}}_{2}$ interface as the microscopic mechanism responsible for HCD. Such a rupture reaction can be driven by a solitary highly energetical carrier; this process is referred to as the single-carrier (SC) mechanism of bond breakage [68,69]. In other words, the model is consistent with the energy-driven paradigm described by Rauch, La Rosa, and Guarin [19,21]. Alternatively, a series of colder carriers can induce the multivibrational excitation of the bond, which results in its weakening and finally rupture; this scenario is referred to as the multiple-carrier (MC) mechanism of bond rupture, as proposed by the group of Hess [4,69,79,80]. In the case of ultra-scaled MOSFETs, it was shown that the most probable pathway of bond dissociation is via coupled MC- and SC-processes [8,9]. Therefore, to calculate the rates of the SC- and MC-mechanisms, one needs to solve the carrier transport sub-task of the entire problem of HCD modeling and obtain the energy distribution function (DF) for carriers. Throughout the paper, we consider HCD in an nMOSFET, which is sketched in Figure 4.

### 3.1. Transport of Primary Carriers

In the TCAD version of our HCD model [9], carrier DFs were obtained by solving the carrier BTE using the deterministic solver ViennaSHE [81,82,83,84], which solved BTE using the expansion of the carrier energy distribution function into a series of spherical harmonics [85,86,87]. Such a solution is computationally expensive, and therefore, in the CPM for both types of carriers, we use the analytical expression for the carrier DF f(E) (with *E* being carrier energy), as proposed by Grasser et al. [88]:(1)$$f\left(E\right)=A\exp\left[-{\left(\frac{E}{E_{\mathrm{ref}}}\right)}^{b}\right]+C\exp\left[-\frac{E}{k_{B}T_{L}}\right],$$
where the first term represents the fraction of non-equilibrium (hot) carriers and the second term corresponds to the Maxwellian distribution of thermalized (cold) carriers; $E_{\mathrm{ref}}$ is the reference energy for hot carriers; *A* and *C* are weighting factors; $k_{B}$ the Boltzmann constant; and $T_{L}$ is the lattice temperature. The exponent *b* is chosen to be 1 within the source and drain regions and 2 elsewhere.

The transistor is represented by a series of slices in the source-drain direction. For each of the slices, we need to obtain the three adjustable parameters $E_{\mathrm{ref}}$, *A*, and *C*, which determine the carrier DF [22]. For primary carriers, in each slice labeled with an index *i*, we solve the system of three equations that are based on three moments of the BTE, with the closures being the carrier concentration (*n*), average carrier energy ($E_{e}$), and the kurtosis ($\beta_{e}$): (2)$$\begin{array}{cc} \; & \int_{0}^{\infty }f_{e,i}\left(E\right)g_{c}\left(E\right)dE=n_{i} \end{array}$$(3)$$\begin{array}{cc} \; & \frac{1}{n_{i}}\int_{0}^{\infty }Ef_{e,i}\left(E\right)g_{c}\left(E\right)dE=E_{e,i} \end{array}$$(4)$$\begin{array}{cc} \; & \frac{3}{5}\;\frac{\langle E^{2}\rangle }{{\langle E\rangle }^{2}}=\frac{3}{5}\;\frac{n_{i}\;\int_{0}^{\infty }E^{2}f_{e,i}\left(E\right)g_{c}\left(E\right)dE}{{\left(\int_{0}^{\infty }Ef_{e,i}\left(E\right)g_{c}\left(E\right)dE\right)}^{2}}=\beta_{e,i}.\; \end{array}$$

For the density-of-states in the Si conduction band $g_{c}\left(E\right)$ in (Equation 2) and (3), we use the analytical expression based on the Kane non-parabolic dispersion relation [88].

The carrier concentration is computed using the Poisson solver implemented in the reliability simulator, Comphy. To calculate the carrier kurtosis, we use the empirical expression proposed by Grasser et al. [88], derived from the rigorous BTE solution (for more detail, see [22]). Within the previous version of our CPM [22], we calculated average electron energy ($E_{e}$) as
(5)$$E_{e}=\frac{3}{2}k_{B}T_{L}+q\tau_{e,E}\mu_{e}F_{\mathrm{Si}}^{2},$$
where $\tau_{e,E}$ is the electron energy relaxation time, $\mu_{e}$ the electron mobility, $F_{\mathrm{Si}}$ the electric field in the channel, and *q* the elementary charge.

Equation (Equation 5) corresponds to the manner of carrier energy evaluation within the DD approach to the BTE solution, which leads to spurious results for short-channel FETs. Figure 5 shows $E_{e}$ plotted as a function of the coordinate *x* along the $\mathrm{Si}/{\mathrm{SiO}}_{2}$ interface evaluated using (Equation 5) and the $E_{e}\left(x\right)$ profile obtained from the rigorous BTE solution with ViennaSHE. These $E_{e}\left(x\right)$ dependencies were obtained for an nMOSFET with $L_{g}$ = 28 nm (the source is at *x* = 0 nm); see Section 2; the applied voltages are $V_{\mathrm{gs}}$ = $V_{\mathrm{ds}}$ = 1.8 V. One can see that these two profiles have different shapes. Even more, at a moderate $V_{\mathrm{gs}}$ of 1.8 V, the *average* electron energy obtained within the DD-based approach reaches a value of ∼10 eV, which is unphysical. Therefore, in the refined CPM for HCD, the carrier transport description needs to be revised.

Within the refined transport modeling approach (Figure 6), we consider two competing mechanisms, i.e., carrier acceleration by the electric field and energy loss due to scattering. Energy gained by carriers is determined by the band bending profile in the source-drain direction, and this profile is obtained from the Poisson solver of Comphy. For compact physics treatment of scattering mechanisms, we assume that a carrier loses an amount of energy δE (=28 meV) each time it passes a distance equal to its mean free path λ. In the simplified version of the model, we use λ=3 nm; this value is consistent with the electron mean free path reported in [89].

Let us comment on the choice of the parameters $\lambda_{0}$ and δE. The carrier mean free path is determined by the scattering rate and the carrier velocity. These two quantities are a function of energy, and therefore, for a thorough evaluation of the mean free path, we need to consider the actual energy DF of the carrier ensemble. Based on our experience in modeling carrier transport in sub-100 nm MOSFETs, we envisage that DFs of substantially hot carriers feature a plateau (i.e., in this local energy range, the DF is a weak function of carrier energy) spreading up to energies of ∼$|qV_{\mathrm{ds}}|$ [78,90]. HCD measurements in modern scaled MOSFETs are conducted at $V_{\mathrm{ds}}$ lying in the range of [1.0,2.0]V. In this energy range, the electron-phonon scattering rate is ∼(0.5–1.0)$\;\times \;10^{14}\;$s${}^{-1}$ [91,92,93]. The hole-phonon scattering rates in the valence band also have comparable values [93]. As for scattering at ionized impurities, Qiu et al. showed that this mechanism is dominant at carrier energies not exceeding ∼0.3eV, while at higher energies, electron-phonon interactions prevail [94]. Therefore, for our estimation, ionized impurity scattering can be neglected. The hole and electron velocities in the aforementioned energy segment are in the range of ∼(0.5–1.0)$\;\times \;10^{8}\;$cm/s [91,95]. By combining the given scattering rates and carrier velocities, we obtain the carrier mean free path to be within an interval of 2–10 nm, i.e., the value λ=3 nm used in our CPM is consistent with our estimation.

The values of energy loss due to the interaction of an electron with an optical phonon are 62.0 and 58.6 meV for longitudinal and transverse optical modes, respectively [95]. For acoustic phonons, these values are 12.1 and 19.0 meV for the two branches of transverse acoustic phonons and 18.4 and 47.4 meV for the longitudinal acoustic phonon branches [95]. Based on the listed values, scattering at acoustic phonons is often considered quasi-elastic, and the corresponding contribution to carrier energy loss is hence neglected. However, recently, Fischetti et al. [92] suggested that this assumption should be revised because, although energy loss due to optical phonon scattering is higher than that typical for scattering at acoustic phonons, the rate of the former mechanism is significantly lower than in the latter case. This idea is consistent with previously published data [96]. Therefore, our energy loss parameter δE=28meV is a reasonable trade-off between energy loss values typical for optical and acoustic phonons.

Figure 7 compares the $E_{e}\left(x\right)$ profiles calculated for $V_{\mathrm{gs}}$ = 1.0 V and $V_{\mathrm{ds}}$ = 2.1 V using the refined carrier transport model and the DD-based approach of Equation (Equation 5). One can see that the former profile is quantitatively similar to that obtained using ViennaSHE (Figure 5) and the maximum $E_{e}$ is ∼1.7 eV, i.e., reasonable for $V_{\mathrm{ds}}$ = 2.1 V. Quite to the contrary, the profile evaluated with the DD-based approach reaches an energy of more than 20 eV, thereby manifesting the inapplicability of Formula (Equation 5).

With $E_{e}$ calculated for each transistor slice, we solve the system (Equation 2)–(4) and obtain the electron energy DF $f_{e}$. An example of generalized electron DFs (i.e., $f_{e,i}\left(E\right)g_{c}\left(E\right)$ with dimensionality of J${}^{-1}$m${}^{-3}$) for $V_{\mathrm{gs}}$ = 1.0 V and $V_{\mathrm{ds}}$ = 2.1 V is shown in Figure 8 for five different positions at the $\mathrm{Si}/{\mathrm{SiO}}_{2}$ interface with *x* = 0.2, 10.2, 20.2, 25.0, and 26.5 nm. The position *x* = 0.2 nm corresponds to the source area, where electrons are thermalized, and therefore the DF is Maxwellian. As *x* changes towards the drain, the DFs shift from equilibrium, which is manifested by the extension of the plateau (with DF values being almost unchanged with increasing *E*). For example, at *x* = 26.5 nm, when almost the entire gate voltage $V_{\mathrm{ds}}$ drops across the channel, this plateau propagates up to ∼1.9 eV. This DF transformation is consistent with the behavior of electron DFs obtained using ViennaSHE; see [9].

### 3.2. Transport of Secondary Carriers

With the obtained DFs for primary electrons, we proceed to the modeling of carrier transport for secondary holes. Secondary carriers are generated by impact ionization, and to evaluate the II rate ($G_{\mathrm{II}}$), we use the model by Grasser et al. [88]:(6)$$G_{\mathrm{II}}=\int P_{\mathrm{II}}\left(E\right)f_{e}\left(E\right)g_{c}\left(E\right)dE,$$
where the reaction rate $P_{\mathrm{II}}\left(E\right)$ is
(7)$$P_{\mathrm{II}}\left(E\right)=P_{0}{\left(\frac{E-E_{\mathrm{th}}}{E_{\mathrm{th}}}\right)}^{2}$$
with $E_{\mathrm{th}}$ = 1.12 eV, i.e., equal to the band gap of Si and $P_{0}$ = 4.18×10${}^{12}\;$s${}^{-1}$.

A comparison of the $G_{\mathrm{II}}$ dependencies calculated with the refined CPM and the DD-based approach is given in Figure 9. Whereas the former profile is in good qualitative agreement with the $G_{\mathrm{II}}$ results from [88], the latter one substantially deviates from them in terms of the $G_{\mathrm{II}}\left(x\right)$ shape and peak values. Therefore, using the $G_{\mathrm{II}}$ rate obtained with the DD based approach would result in a severely overestimated contribution of secondary holes and hence spurious $\Delta I_{d,\mathrm{lin}}$ values. Figure 10 provides a summary of $G_{\mathrm{II}}\left(x\right)$ profiles evaluated with different values of energy loss: δE = 28, 35, and 42 meV. The increasing value of δE results in a lower average energy of the electron ensemble and therefore a smaller rate $G_{\mathrm{II}}$.

For secondary carriers, which are generated by impact ionization, we do not have access to the hole concentration (*p*), and therefore, instead of the system (Equation 2)–(4), we employ a modified set of equations: (8)$$\begin{array}{cc} \; & \frac{1}{p_{i}}\int_{0}^{\infty }Ef_{h,i}\left(E\right)g_{v}\left(E\right)dE=E_{h,i} \end{array}$$(9)$$\begin{array}{cc} \; & \frac{3}{5}\;\frac{\langle E^{2}\rangle }{{\langle E\rangle }^{2}}=\frac{3}{5}\;\frac{p_{i}\;\int_{0}^{\infty }E^{2}f_{h,i}\left(E\right)g_{v}\left(E\right)dE}{{\left(\int_{0}^{\infty }Ef_{h,i}\left(E\right)g_{v}\left(E\right)dE\right)}^{2}}=\beta_{h,i} \end{array}$$(10)$$\begin{array}{cc} \; & J_{h,i}^{\mathrm{out}}=J_{h,i-1}^{\mathrm{in}}+G_{\mathrm{II},i}l_{s}-R_{i}l_{s}. \end{array}$$

In this system, Equation (10) is the flux balance equation for holes. For each slice *i*, we assume that the supply of secondary holes should be equal to the loss of holes; see Figure 11. The supply components are hole generation by II, designated as $G_{\mathrm{II},i}l_{s}$ ($l_{s}$ is the slice length), and the hole flux $J_{h,i-1}^{\mathrm{in}}$ from the previous slice with index i−1. Note that in the case of holes, the slice enumeration begins at the drain (this slice has *i* = 0), and the index *i* increases towards the source (Figure 11). Hole loss is due to recombination with the rate $R_{i}$ and the flux $J_{h,i}^{\mathrm{out}}$ of holes departing from the slice *i* to the slice i+1. The flux of holes entering slice *i* (see Figure 11) is calculated as
(11)$$J_{h,i-1}^{\mathrm{in}}=\frac{2}{\pi }\;\int f_{h,i-1}\left(E\right)g_{v}\left(E\right)v_{h,i-1}\left(E\right)dE,$$
where $f_{h,i}$ is the hole energy DF, $g_{v}$ is the density of states in the valence band, and $v_{h,i}$ is the velocity of holes. The coefficient 2/π is related to the averaging of cosθ (where θ is the angle between the carrier velocity and the transport direction) over a uniform distribution of θ∈[−π/2;π/2]. Holes leaving the slice *i* (flux $J_{h,i}^{\mathrm{out}}$ in Figure 11) can move in any direction and therefore the coefficient 2/π is omitted:(12)$$J_{h,i-1}^{\mathrm{out}}=\int f_{h,i-1}\left(E\right)g_{v}\left(E\right)v_{h,i-1}\left(E\right)dE,$$

The hole concentration *p* enters (9) and it is evaluated as
(13)$$p_{i}=\int f_{h,i}\left(E\right)g_{v}\left(E\right)dE.$$

For the recombination rate $R_{i}$ we assume that the concentration of secondary holes is much less than that of primary electrons, i.e., p≪n, and therefore [97]:(14)$$R_{i}=p_{i}/\tau_{E,h},$$
where $\tau_{E,h}$ is the energy relaxation time for holes.

Hole DFs obtained by solving the system of Equations (Equation 8)–(10) for $V_{\mathrm{gs}}$ = 1.0 V and $V_{\mathrm{ds}}$ = 2.1 V are plotted in Figure 12 for different positions along the interface. One can see that for the drain area at *x* = 26 nm (this *x* value corresponds to the II rate peak, Figure 10), where holes are predominantly generated, the DF is Maxwellian because holes are thermalized. However, holes are accelerated by the electric field towards the source and their DFs become strongly non-equilibrium. The impact of the carrier energy loss δE on hole DFs is depicted in Figure 13 for two positions along the interface; like in Figure 10, we used δE = 28, 35, and 42 meV. One can see that the increasing δE leads to lower values of DFs in the entire energy range. This trend appears to be very reasonable because a higher δE results in a lower II rate (Figure 10), thereby decreasing the hole concentration, and holes themselves lose more energy, i.e., become colder. Finally, Figure 14 provides a comparison of hole DFs calculated with the refined carrier transport treatment and within the DD-based approach. It can be seen that the latter DFs have enormously high values, and their behavior is consistent with the spurious II rate (Figure 9) obtained using (Equation 5).

The evaluated DFs for both types of carriers are then used to calculate bond dissociation rates and the interface trap density $N_{\mathrm{it}}$ as a function of the lateral coordinate *x* for each stress time step *t*; for details, see [9,78]. The $N_{\mathrm{it}}\left(x\right)$ profile is then employed to calculate $\Delta I_{d,\mathrm{lin}}\left(t\right)$ traces taking into account both electrostatic perturbation of the stressed device and mobility reduction; this procedure is described in [22].

It is important to emphasize that the developed CPM for HCD allows one to dramatically reduce computational time. The most computationally expensive part of our TCAD model for HCD is transport simulation. Depending on the device architecture (its complexity, the number of mesh points, etc.) and stress conditions, solving the BTE for a real device structure may require a few hours. For example, transport simulations carried out for the MOSFET employed in this study took approximately 3 h on a desktop. As for the CPM, its accuracy depends on the number of slices used to reproduce the transistor. However, increasing the number of slices would increase computational time (roughly) proportionally. Thus, it is important to find a balance between the number of slices (computational time) and model accuracy. For the calculations presented in this work, we used a relatively large number of slices, namely 100. As the gate length of our devices is 28 nm, using 100 slices results in a good resolution comparable to that provided by fine meshes used in commercial device simulators. In this case all calculations (including transport modeling, calculations of the $N_{\mathrm{it}}$ density, and finally obtaining $\Delta I_{d,\mathrm{lin}}\left(t\right)$ traces) were completed within 1–2 min on a laptop.

## 4. Degradation Characteristics

For the case of the HCD WCC in short-channel transistors with $V_{\mathrm{gs}}$ = $V_{\mathrm{ds}}$ = 1.8, 1.9, and 2.0 V, $\Delta I_{d,\mathrm{lin}}\left(t\right)$ traces are summarized in Figure 1. One can see that the refined model can capture experimental data with good accuracy. $\Delta I_{d,\mathrm{lin}}\left(t\right)$ curves modeled with increased values of the energy loss parameter δE of 35 and 42 meV have lower values than those simulated with δE = 28 meV. This tendency is consistent with the impact of δE on the II rate $G_{\mathrm{II}}$ (Figure 10) and hole DFs (Figure 13). We also simulated $\Delta I_{d,\mathrm{lin}}\left(t\right)$ dependencies disregarding the contribution of secondary holes, and one can see that these traces coincide with those obtained using the “full” model. In other words, if we neglect the HCD component driven by holes, we do not underestimate $\Delta I_{d,\mathrm{lin}}$ changes, i.e., at $V_{\mathrm{gs}}$ = $V_{\mathrm{ds}}$, the impact of secondary holes is not significant.

Quite to the contrary, at a much lower $V_{\mathrm{gs}}$ of 1.0 V, neglecting the contribution of secondary holes results in substantial underestimation of HCD (see Figure 2). The same behavior is also pronounced at $V_{\mathrm{gs}}$ = 0.69 V (Figure 3), but in a less prominent way. Such a trend can be understood considering that the secondary holes are generated by II, whose rate features a maximum at $V_{\mathrm{gs}}$ = (0.4–0.5)$V_{\mathrm{ds}}$ [97]. This interrelation of the voltages corresponds to the stress conditions with $V_{\mathrm{gs}}$ = 1.0 V and, to a lesser extent, to the regimes with $V_{\mathrm{gs}}$ = 0.69 V being in mismatch with the WCC.

Another important peculiarity noticeable in Figure 2 and Figure 3 is that the contribution of secondary holes becomes more significant at higher $V_{\mathrm{ds}}$. This is because with an increasing $V_{\mathrm{ds}}$, primary carriers become hotter, thereby resulting in a higher II rate and a higher concentration of secondary carriers; the secondary carriers also reach higher energies at higher $V_{\mathrm{ds}}$. This trend is confirmed by $N_{\mathrm{it}}\left(x\right)$ profiles obtained with and without the secondary hole contribution for $V_{\mathrm{gs}}$ = 1.0 V and $V_{\mathrm{ds}}$ = 1.9 and 2.1 V, see Figure 15. From Figure 15, we conclude that, in addition to the “traditional” $N_{\mathrm{it}}$ peak located at the drain and originating from primary carriers [98,99,100], secondary holes result in an $N_{\mathrm{it}}$ peak situated near the source, whose position is consistent with the results obtained using the TCAD model based on the rigorous BTE solution [30,101].

It is noteworthy that although the contribution to HCD provided by secondary holes is most prominent at the highest $V_{\mathrm{ds}}$, it results in a change in the time slope of modeled $\Delta I_{d,\mathrm{lin}}\left(t\right)$ traces. The data set used in this study was acquired within the time window limited by 144 s; however, at longer stress times (e.g., several ks), deviations of $\Delta I_{d,\mathrm{lin}}$ values calculated disregarding the impact of secondary carriers from experimental $\Delta I_{d,\mathrm{lin}}$ changes would be quite substantial. On the other hand, the major task the model aims at tackling is to—based on experimental data acquired at high stress voltages—predict device time-to-failure for regimes with operating voltages comparable to $V_{\mathrm{dd}}$. Even though $V_{\mathrm{dd}}$ in the employed MOSFETs is 1.2 V, and therefore, in the aforementioned regimes, the impact of secondary carriers is weak, the model validation/calibration disregarding their contribution would result in a spurious value of device lifetime.

Let us emphasize that the model can thoroughly reproduce experimental $\Delta I_{d,\mathrm{lin}}\left(t\right)$ traces for all stress conditions.

## 5. Conclusions

We extended our compact physics model for hot-carrier degradation by implementing the component driven by secondary carriers generated by impact ionization. This implementation is based on refined carrier transport modeling for both types of carriers. Note that in the previous version of our CPM for HCD, the average carrier energy was estimated using the homogeneous energy balance equation, but this drift-diffusion-based treatment has very limited applicability in short-channel FETs. In the extended CPM, carrier energy is evaluated taking into account the band bending profile in the transport direction and the energy dissipation parameters such as the carrier mean free path and energy loss due to scattering.

The extended CPM was validated against HCD data acquired from foundry-quality nFETs (with secondary carriers being holes). Within model validation, the contribution of secondary holes was shown to be weak at the worst-case conditions for HCD ($V_{\mathrm{gs}}$ = $V_{\mathrm{ds}}$) but became very significant at lower $V_{\mathrm{gs}}$. This trend stems from two reasons: (1) the II rate, which generates the secondary carriers, is at its maximum when $V_{\mathrm{gs}}∼0.5\;V_{\mathrm{ds}}$ (which is shifted from the WCC), and (2) under the WCC, the secondary hole contribution is screened by the damage generated by primary electrons. It has also been shown that the hole-induced portion of HCD becomes stronger at higher $V_{\mathrm{ds}}$ values. This is because at a higher $V_{\mathrm{ds}}$ primary carriers have higher energies, thereby resulting in a higher II rate and a higher concentration of generated electron-hole pairs; in addition, secondary holes themselves can reach higher energies under an increased $V_{\mathrm{ds}}$. Note that the interface trap density peak caused by secondary holes is located near the source, which is consistent with the results obtained with the full TCAD version of our HCD model. Finally, the extended CPM model was shown to accurately capture $\Delta I_{d,\mathrm{lin}}\left(t\right)$ traces over a broad range of stress conditions.

## Figures and Tables

**Figure 1 micromachines-14-02018-f001:**
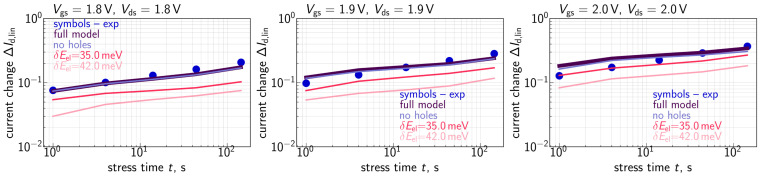
Experimental and calculated $\Delta I_{d,\mathrm{lin}}\left(t\right)$ traces for the WCC of HCD with $V_{\mathrm{gs}}$ = $V_{\mathrm{ds}}$ = 1.8, 1.9, and 2.0 V. To analyze the role of secondary holes also $\Delta I_{d,\mathrm{lin}}\left(t\right)$ traces obtained without their contribution were evaluated. We use the energy loss parameter of δE = 28 meV, but the model results are very sensitive to its variations (see Figures 10 and 13 for the impact of the energy loss parameter) and therefore $\Delta I_{d,\mathrm{lin}}\left(t\right)$ curves for δE = 35 and 43 meV are also shown.

**Figure 2 micromachines-14-02018-f002:**
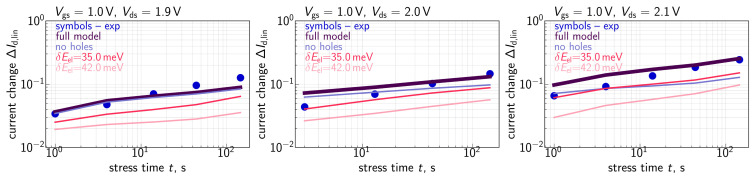
The same as in Figure 1 but for $V_{\mathrm{gs}}$ = 1.0 V and $V_{\mathrm{ds}}$ = 1.9, 2.0, and 2.1 V.

**Figure 3 micromachines-14-02018-f003:**
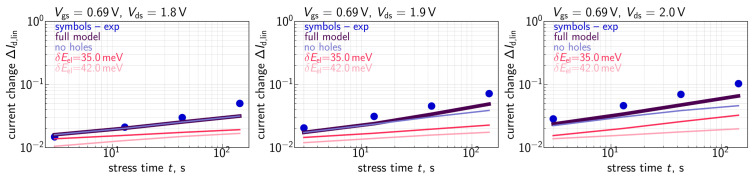
The same as in Figure 1 but for $V_{\mathrm{gs}}$ = 0.69 V and $V_{\mathrm{ds}}$ = 1.8, 1.9, and 2.0 V.

**Figure 4 micromachines-14-02018-f004:**
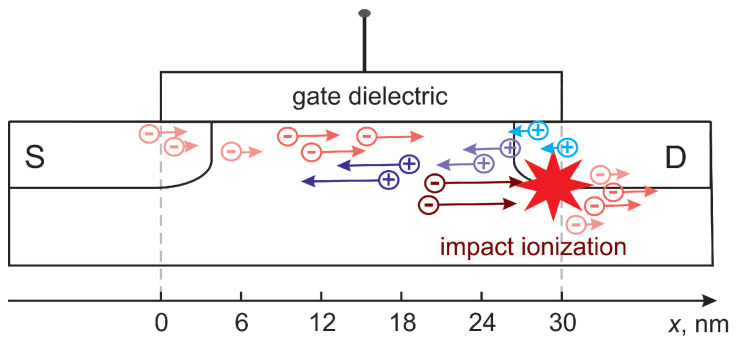
Schematic representation of the device used to validate the model. The device source is at x=0. Primary carriers, which drive impact ionization, are electrons. Secondary holes generated by impact ionization near the drain are accelerated towards the source where they reach highest energies and provide the most significant contribution to HCD.

**Figure 5 micromachines-14-02018-f005:**
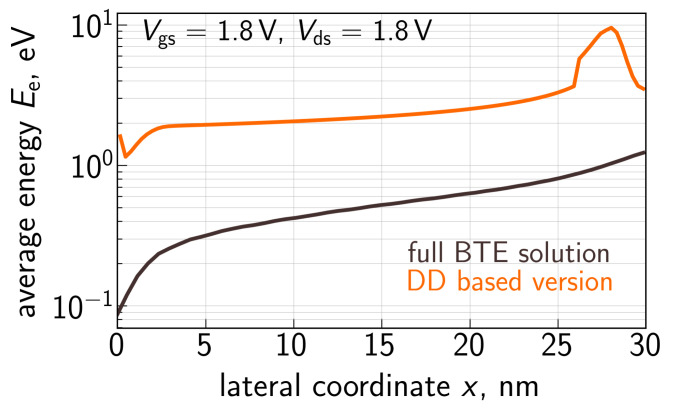
Average energy of primary electrons $E_{e}$ as a function of the lateral coordinate *x* along the $\mathrm{Si}/{\mathrm{SiO}}_{2}$ interface of an n-channel MOSFET with a gate length of 28 nm (*x* = 0 nm corresponds to the source). Shown are two $E_{e}\left(x\right)$ profiles: one obtained with the DD-based approach and another one calculated employing the BTE solution with the carrier transport simulator ViennaSHE. This comparison illustrates that the DD based approach to estimation of average carrier energy, and further to HCD modeling, is not applicable.

**Figure 6 micromachines-14-02018-f006:**
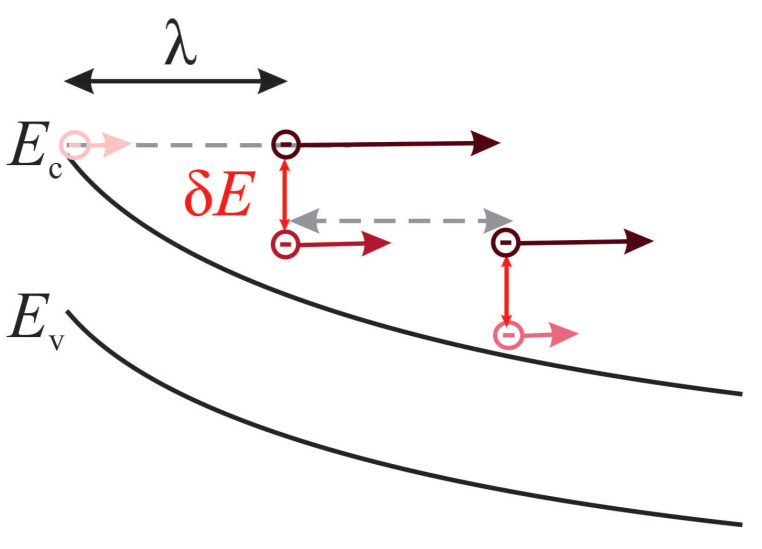
In the refined CPM for HCD, the average energy of electrons is determined by carrier acceleration by the electric field and energy dissipation due to scattering. Energy gained from the electric field is evaluated based on the band bending profile in the transport direction. We assume that when an electron travels a distance equal to its mean free path (λ) it loses a certain amount of energy δE.

**Figure 7 micromachines-14-02018-f007:**
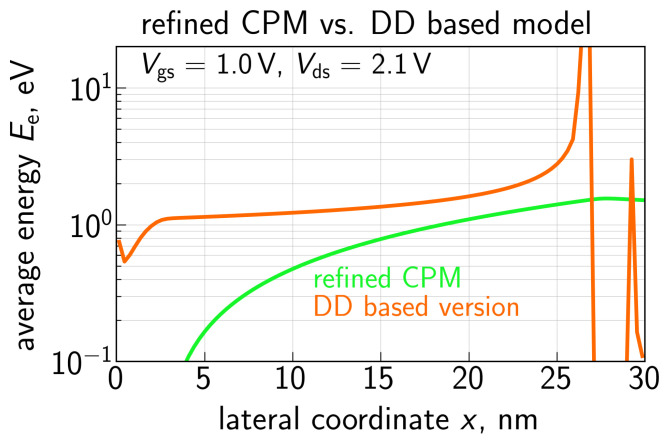
Average energy of primary electrons vs. the lateral coordinate *x* obtained with the refined CPM and compared with that evaluated using the DD based approach.

**Figure 8 micromachines-14-02018-f008:**
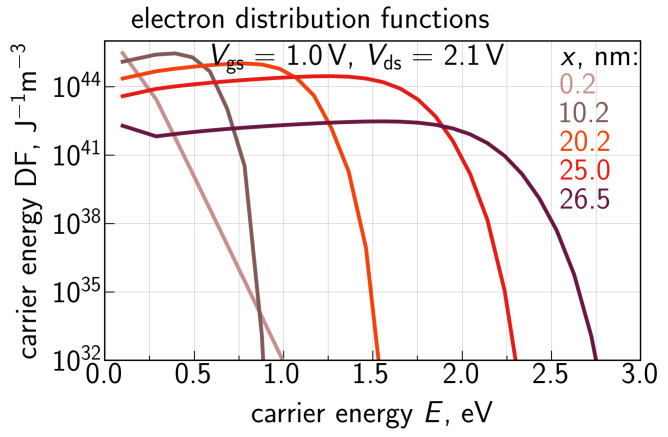
Generalized distribution functions of primary electrons obtained with the refined CPM for different positions along the $\mathrm{Si}/{\mathrm{SiO}}_{2}$ interface.

**Figure 9 micromachines-14-02018-f009:**
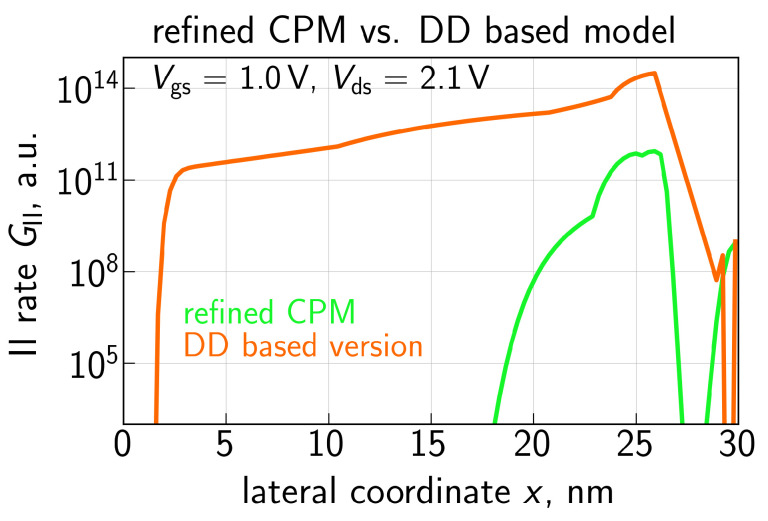
The impact ionization rate $G_{\mathrm{II}}$ as a function of *x* calculated with the refined CPM and the DD based formula (the latter approach overestimates $G_{\mathrm{II}}$).

**Figure 10 micromachines-14-02018-f010:**
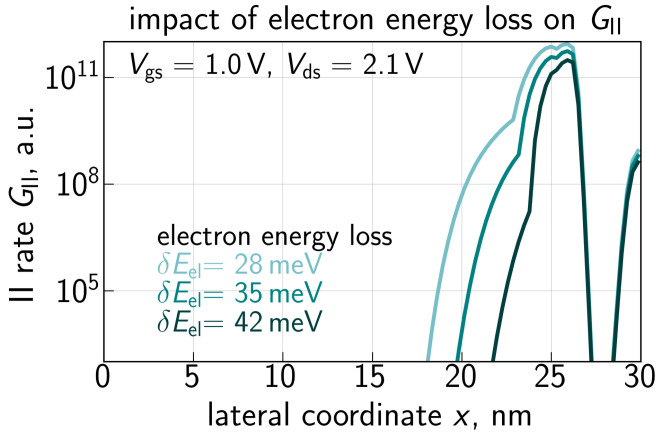
The II rate $G_{\mathrm{II}}$ vs. the coordinate *x* obtained for three different values of the energy loss parameter δE = 28, 35, and 42 meV.

**Figure 11 micromachines-14-02018-f011:**
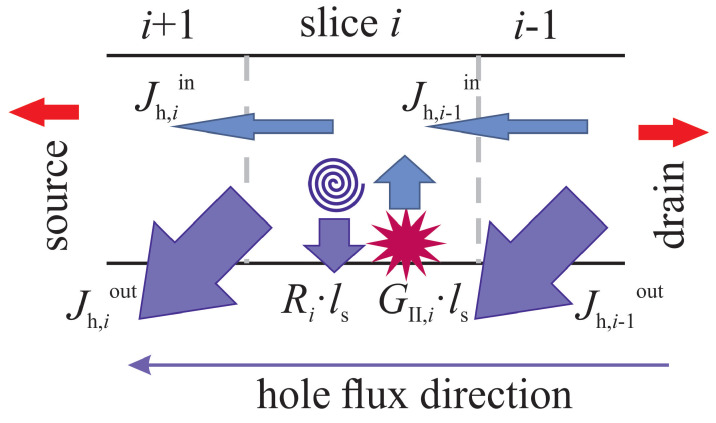
Schematic representation of the hole flux balance equation. For each transistor slice with an index *i* (for holes the slice enumeration begins at the drain, where the holes are predominantly created by II, i.e., *i* increases towards the source) a balance between hole supply and hole loss is considered. The hole supply component is due to holes arriving from the previous slice i−1 (the hole flux $J_{h,i-1}^{\mathrm{in}}$) and impact ionization (with the rate $G_{\mathrm{II}}$), while hole loss is due to hole departure to the slice i+1 ($J_{h,i}^{\mathrm{out}}$) and recombination (with the rate *R*). The slice length is designated as $l_{s}$.

**Figure 12 micromachines-14-02018-f012:**
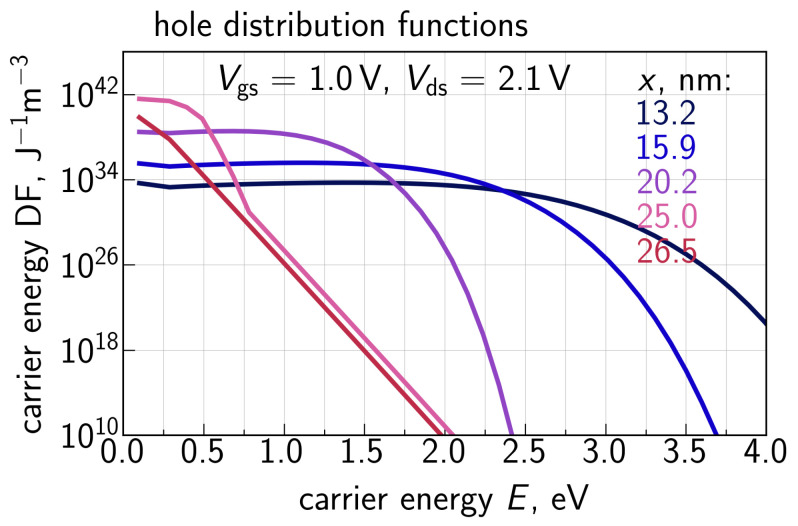
Generalized DFs for secondary holes calculated with the refined compact physics model and plotted for different positions at the $\mathrm{Si}/{\mathrm{SiO}}_{2}$ interface.

**Figure 13 micromachines-14-02018-f013:**
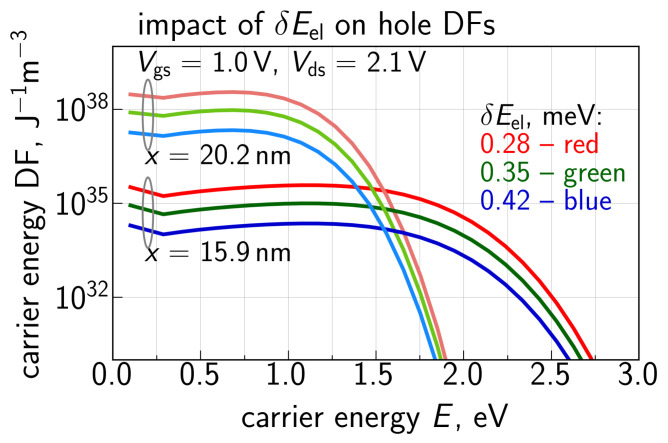
Impact of the energy loss parameter δE on the secondary hole DFs. At larger δE both types of carriers are colder and this trend is confirmed by the hole DFs.

**Figure 14 micromachines-14-02018-f014:**
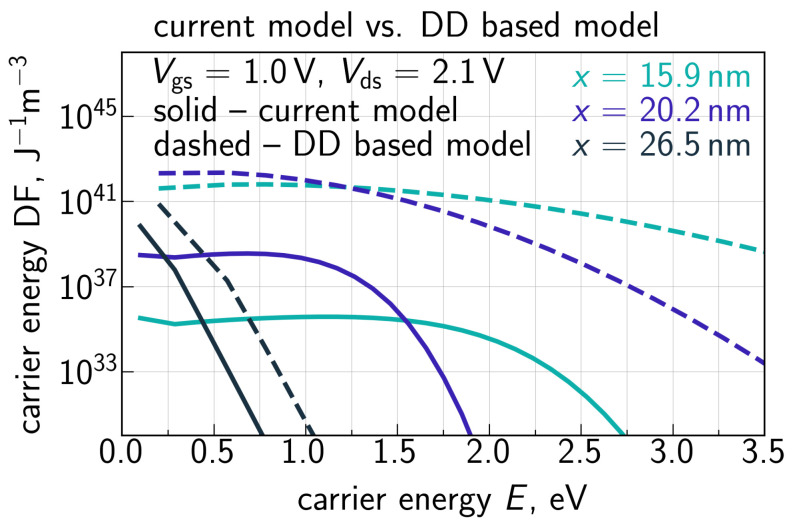
Comparison of hole DFs obtained with refined carrier transport treatment and using the DD based model. In the latter case the DFs have spuriously high values.

**Figure 15 micromachines-14-02018-f015:**
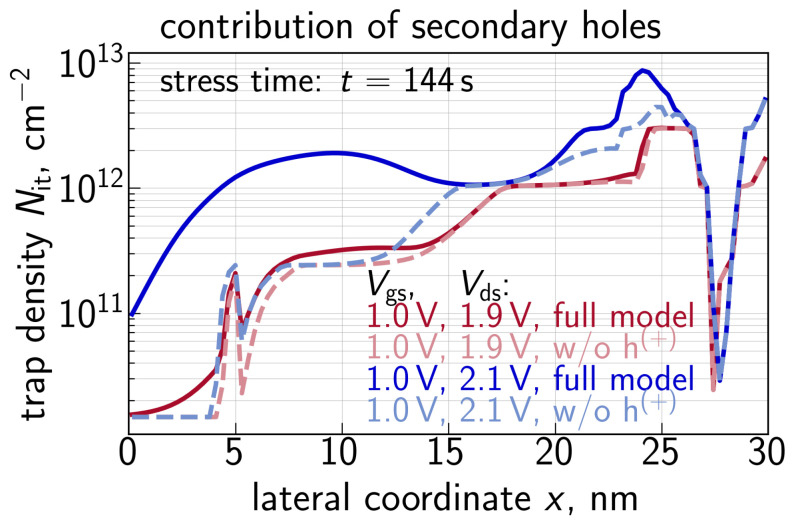
The interface trap density $N_{\mathrm{it}}$ as a function of the coordinate along the interface *x* for $V_{\mathrm{gs}}$ = 1.0 V and two values of $V_{\mathrm{ds}}$ = 1.9 and 2.1 V calculated with and without the contribution of secondary holes. One can see that the hole contribution results in the secondary $N_{\mathrm{it}}$ peak situated near the source, and this peak becomes more pronounced at higher $V_{\mathrm{ds}}$.

## Data Availability

The data presented in this study are available on request from the corresponding author.

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
