# Peer review of "Compact Physics Hot-Carrier Degradation Model Valid over a Wide Bias Range"

_micromachines, 2023, doi:10.3390/mi14112018_

Round 1

Reviewer 1 Report

Comments and Suggestions for Authors

The paper deals with the study of the secondary carriers in the hot-carrier degradation of 28nm n-channel MOSFETs. The description of the modeling approach is compleate and clear. There are some parts of the manuscript that need some improvement for a full understanding of the approach and of the reported outcomes. The first comment is on the choice of experiments from 1sec to 144 sec. This range is quite limited: all experiments show drifts significantly larger than 1% , even larger that 10% at 1 sec, thus experimnets are already in a quite saturated condition, while it would be interesting to check the characteristics in the faster time range. Otherwise, the comments on the exponents typical of the drain-current drifts might give rise to misleading assumptions. Moreover, are the experiments dealing with permanent drifts or recovery is partially expected?

On the model description: in Figures 7,8, 9, 10 the full BTE solution might be added to fully comment on the validation of the proposed approach. How relevant is the effect of the analytical density-of-states in Eqs. 2, 3 and 4? It would be very interesting also to comment on the advantage in ters of simulation time when comparing the full-band BTE solution to the proposed approach. 

As far as Eq. (6) is concerned, is it related to the Coulomb effect? It is not clear why the doping concentration is such relevant in the empirical formula in (7), while the DFs are stated to be mostly influenced by the electron-phonon scattering (as expected). 

On the results reported in Fig. 12: why the first DF is smaller than the second one in position 25 nm if it is the position of the II-rate peak?  Are the main outcomes reported in comparison with experiments confirmed also by the accurate full-band approach? 

Author Response

A detailed reply to the individual comments is uploaded as a separate document.

Reviewer 2 Report

Comments and Suggestions for Authors

This work lays out how to efficiently model DF for simulation of II in a consistent framework. The concept is well presented and appears meaningful under the assumption that the cited work is correct.
This might be also the weak point as there is not much experimental validation, still, the authors do include some (albeit limted short term) measurement. In light of a simulation and theory heavy paper this is deemed acceptable.
Maybe a note about the measurement delay could be added, and since there is plenty of statistical data, it could be interesting to show/discuss this briefly. In praticular as how analysis of the single devices might aid understanding or corroborate the model.
For the simulation method using slices, an obvious question may be on how the distance between the slices affects the simulations. Adding a note on this might be interesting for some readers.

Author Response

(The authors gave the same response as above.)

Round 2

Reviewer 1 Report

Comments and Suggestions for Authors

The Authors answered to all my comments in a clear way. The manuscript can be published as is.